# Multimorbidity and Complex Multimorbidity in India: Findings from the 2017–2018 Longitudinal Ageing Study in India (LASI)

**DOI:** 10.3390/ijerph19159091

**Published:** 2022-07-26

**Authors:** Abhinav Sinha, Sushmita Kerketta, Shishirendu Ghosal, Srikanta Kanungo, John Tayu Lee, Sanghamitra Pati

**Affiliations:** 1ICMR-Regional Medical Research Centre, Bhubaneswar 751023, India; dr.abhinav17@gmail.com (A.S.); sushmita.kerketta07@gmail.com (S.K.); shishirendu123@gmail.com (S.G.); 2The Nossal Institute for Global Health, Melbourne School of Population and Global Health, The University of Melbourne, Melbourne, VIC 3010, Australia; johntayulee@unimelb.edu.au; 3Public Health Policy Evaluation Unit, Department of Primary Care and Public Health, School of Public Health, Imperial College London, London SW7 2AZ, UK

**Keywords:** multimorbidity, complex multimorbidity, ageing, LASI, India

## Abstract

Complex multimorbidity refers to the co-occurrence of three or more chronic illnesses across >2 body systems, which may identify persons in need of additional medical support and treatment. There is a scarcity of evidence on the differences in patient outcomes between non-complex (≥2 conditions) and complex multimorbidity groups. We evaluated the prevalence and patient outcomes of complex multimorbidity and compared them to non-complex multimorbidity. We included 30,489 multimorbid individuals aged ≥45 years from the Longitudinal Ageing Study in India (LASI) from wave-1 conducted in 2017–2018. We employed a log link in generalised linear models (GLM) to identify possible risk factors presenting the adjusted prevalence–risk ratio (APRR) and adjusted prevalence–risk difference (APRD) with 95% confidence interval. The prevalence of complex multimorbidity was 34.5% among multimorbid individuals. Participants residing in urban areas [APRR: 1.10 (1.02, 1.20)], [APRD: 0.04 (0.006, 0.07)] were more likely to report complex multimorbidity. Participants with complex multimorbidity availed significantly higher inpatient department services and had higher expenditure as compared to the non-complex multimorbidity group. Our findings have major implications for healthcare systems in terms of meeting the requirements of people with complicated multimorbidity, as they have significantly higher inpatient health service utilisation, higher medical costs, and poorer self-rated health.

## 1. Introduction

Multimorbidity is the co-occurrence of two or more long-term conditions in an individual without defining an index disease [1]. Previously, multimorbidity was thought to be more prevalent in high-income countries (HICs), but new research reveals that it is also prevalent in low- and middle-income countries (LMICs) [2]. A recent systematic study on multimorbidity among LMICs found its incidence varying from 0.7 percent to 81.3 percent across different age groups, which can be related to these nations’ epidemiological and demographic transitions [3]. Multimorbidity has become the norm among the adult and older population in India, as it has in other LMICs [4]. A previous study in India estimated the burden of multimorbidity to be over 50% among 45-year or above participants [5].

Despite the widespread use of the term, the traditional method of estimating multimorbidity by counting two or more chronic conditions ignores the heterogeneity of prevalent conditions across the body’s organ systems [6,7]. As a result, there is a growing interest in distinguishing between multimorbidity and complex multimorbidity [8], which is defined as the co-occurrence of three or more chronic conditions affecting more than two body systems in a single individual. When multiple organ systems are involved, new evidence suggests that patients are more vulnerable to emergency admission, unplanned hospitalisation, and polypharmacy [9]. Furthermore, this complicates care-seeking pathways and makes prioritisation difficult for patients [6].

The burden of multimorbidity in India is rapidly increasing due to increased longevity and exposure to chronic disease risk factors [10,11]. However, there is a lack of data on the prevalence and impact of complex multimorbidity in India. There are few studies on complex multimorbidity from high-income countries that focus on population-based prevalence [12,13], and none from India. Identifying individuals with complex multimorbidity among multimorbid people is important because it can help assess the severity and complexity of multimorbidity, allowing for the prioritisation of those who need additional healthcare support and attention. India’s health system traditionally focuses on individual conditions instead of a patient-centred approach [14]. Clinical decision-making for individual conditions is complicated by conflicting clinical guidelines [15]. As a result, evidence on the extent of complex multimorbidity in India is urgently required to guide current healthcare programmes and future policies.

To fill this important evidence gap, this study aims to estimate the prevalence of complex multimorbidity among multimorbid individuals and compare patient outcomes such as healthcare utilisation, healthcare expenditure, and self-rated health (SRH) between individuals with non-complex multimorbidity and complex multimorbidity using nationally representative data from the Longitudinal Ageing Study in India (LASI), wave 1.

## 2. Materials and Methods

### 2.1. Study Design and Participants

The Longitudinal Ageing Study in India (LASI) is a collaborative initiative among the International Institute of Population Sciences (IIPS), Harvard T. H. Chan School of Public Health (HSPH), and the University of Southern California (USC) in the field of ageing population, their health, economic conditions, as well as social behaviour from India. The first wave of LASI was conducted from April 2017 to December 2018 and will be led biennially during the upcoming years. A multistage stratified area probability cluster sampling design was operationalized to make the study population nationally representative. LASI adopted a three-stage sampling design in rural areas and a four-stage sampling design in urban areas to reach the eventual sampling unit. Sub-districts formed the primary sampling unit (PSU) in each state/UT selected randomly from the sampling frame based on the 2011 census. In the second stage, villages from selected sub-districts were chosen randomly, whereas in urban areas, urban wards followed by a census enumeration block (CEB) were selected. After that, households from selected villages/CEB were chosen to reach the target individuals, making LASI nationally representative data (Appendix A). From selected LASI eligible households (LEHs), household heads aged ≥45 years and their spouses (irrespective of age) were administered three survey schedules (viz. community survey schedule; household survey schedule; and individual survey schedule) after receiving written informed consent. With an individual response rate of 87.1%, a total of 72,250 individuals were recruited as a study sample. The detailed study method of LASI has been explained in the LASI India report [16]. The Indian Council of Medical Research (ICMR) (New Delhi) and the IIPS (Mumbai) provided the ethical approval for conducting India’s first wave of panel surveys on the ageing population. Written consent was received for ‘household’ and ‘individual’ before collecting verbal data or blood samples and other parameters.

In India, non-communicable diseases usually appear by around 45 years of age, a decade earlier than many HICs [17]. Hence, we included multimorbid individuals (those with two or more chronic conditions without considering body systems) aged 45 years and above whose biomarker readings were available for this study. After meeting the inclusion and exclusion criteria, the final sample size for this study was 30,489 participants (Figure 1).

### 2.2. Socio-Demographic and Socio-Economic Characteristics

Our study was adjusted for the respondent’s age (categorised into three groups, i.e., 45–59 years; 60–74 years; and 75 years or more); gender (male/female); and place of residence (urban/rural). The castes were grouped as: scheduled tribe, scheduled caste, other “backward” class, and “others”, which was based on two questions: “What is your caste or tribe?”, with options as (1) caste; specify; (2) tribe, specify; (3) no caste/tribe, and “Do you belong to a scheduled caste, a scheduled tribe, other backwards class, or none of these?” Participants’ responses to the second question were directly used to form “scheduled caste”, “scheduled tribes” and “other backward class” groups, whereas “others” comprised of participants who responded with “no caste/tribe” in the first question, along with those who said “none of these” in the second question. Education level (“no formal education”, “up to primary”, “middle school to higher secondary & diploma”, and “graduation & above”) was based on a self-reported highest educational level of education. Employment status was grouped on the basis of past working status for at least three consecutive months in a lifetime with categories formed as “never worked”, “currently working”, and “currently not working”. Wealth index was grouped as poorest, poorer, middle, richer, and richest based on monthly per capita expenditure (MPCE), based on food and non-food expenditures over a 30-day reference period. The reported MPCE of all participants was divided into quintile to form the wealth index. All data were collected through face-to-face participant interviews by a trained LASI team.

Individuals visiting the out-patient department (OPD) or who had been admitted at least once during the last 12 months were sub-grouped from those who never went to a facility. The latest hospitalization charges (in Indian Rupees ₹) were collected for up to four most recent visits, whereas expenses only for the most recent OPD visit were procured under ten different headings. All segregated amounts were summed together to assess each individual’s total OPD or in-patient department (IPD) cost. Participants who availed any OPD service or were admitted for at least one night in an IPD of any hospital were labelled as “at least one”, whereas the rest were marked as “none”. Participants were asked to subjectively rate their health based on the question: “In general, would you say your health is excellent, very good, good, fair, or poor?” Respondent’s self-rated health (SRH) was ranked on a five-point Likert scale as “excellent”, “very good”, “good”, “fair”, or “poor” based on their responses. SRH is widely used as a proxy indicator for health-related quality of life.

### 2.3. Outcome

As the primary outcome of interest, this study assessed two mutually exclusive groups: non-complex multimorbidity and complex multimorbidity. We considered the eighteen most commonly prevalent chronic conditions based on an extensive literature search [18]. These conditions included objectively diagnosed obesity based on body mass index (BMI) and seventeen self-reported chronic conditions: hypertension, diabetes, cancer, chronic lung disease, chronic heart disease, stroke, chronic bone/joint conditions, psychological or neurological conditions, hypercholesterolemia, thyroid disorders, gastrointestinal problems, skin disease, chronic kidney disease, urine incontinence, oral conditions, and vision problems. The self-reported chronic conditions were based on the question: “Has any health professional ever diagnosed you with the following chronic conditions or diseases?” with responses as “yes” or “no”. The participants who responded “yes” were considered to have a chronic condition. Obesity was calculated through BMI considered as weight in kg divided by height in m^2^. Weight (in kg) was collected using a Seca 803 digital weighing scale whereas height (in centimetres) was measured using a stadiometer. We used the World Health Organization’s BMI classification for South Asian adults to define the cut off for obesity, i.e., BMI as 25 kg/m^2^ [19].

Whereas co-existence of two or more conditions without considering body systems was considered as non-complex multimorbidity, complex multimorbidity was defined as three or more chronic conditions among more than two body systems. These chronic conditions were further catalogued into eleven system-specific chapters following the International Classification of Diseases, Tenth Revision (ICD 10) to form complex multimorbidity (Table 1). We aggregated the total number of system-specific chapters individually. Individuals with complex multimorbidity were defined as those who scored at least three out of eleven system-specific chapters. Hence, one or more chronic condition in each chapter will form at least three chronic conditions across three or more system-specific chapter as done in the previous literature [12,20].

### 2.4. Statistical Analysis

The frequency and percentage of multimorbidity and complex multimorbidity were first summarised across socio-demographic groups. The chi-squared test was used to see if there was a statistically significant difference of healthcare utilisation and self-rated health between non-complex and complex multimorbidity groups.

Due to the high prevalence of complex multimorbidity, odds ratios deviate from their actual relative risk, making interpretation difficult. Consequently, the prevalence risk ratio (PRR) and the prevalence risk difference (PRD) were favoured as more precise methods to estimate association. Therefore, we utilised a log link in generalised linear models (GLM) to identify potential risk factors by applying the PRR and 95% CI associated with complex multimorbidity among various socio-demographic characteristics presented as prevalence risk ratio (PRR) and prevalence risk difference (PRD). Finally, we adjusted the models for various socio-demographic correlates among individuals with complex multimorbidity, as expressed by the adjusted prevalence risk ratio (APRR) and the adjusted prevalence risk difference (APRD). A multivariable logistic regression model was performed to find the association between inpatient visits and complex multimorbidity presented as adjusted odds ratio (AOR) with 95% CI. Pattern analysis (dyads and triads) of frequently occurring chronic conditions as well as conditions grouped by ICD-10 chapters were done employing a simple matrix approach that used a comprehensive and exhaustive analysis of all possible combinations of conditions with frequency of more than 1%. We used a median and inter-quartile range (IQR) as a measure to present healthcare expenditure as the data on out-of-pocket expenditure was skewed. The LASI survey weighting (to compensate for complex survey design) was used in our study to ensure national estimates. A statistical analysis was conducted using STATA (StataCorp. 2019. Stata Statistical Software: Release 16. College Station, TX: StataCorp LLC).

## 3. Results

This study included 30,489 adults aged ≥45 years who had two or more chronic conditions. The participants’ average age was 60.8 years. Almost half of the participants (47.2 percent) were between the ages of 45 and 59. In the study population, there was a higher proportion of female in the sample (57.4 percent). The majority of the participants came from rural areas, had no formal education, and were currently employed (Table 2).

Chronic conditions classified as the digestive system (72.34 percent) had the highest prevalence, followed by the circulatory system (51.06 percent) and the musculoskeletal/connective tissue (29.14 percent), with mental/behavioural conditions having the lowest prevalence (Figure 2). Almost one third of the multimorbid individuals (34.51 percent) had complex multimorbidity. The most prevalent chronic condition was hypertension (47.9%) followed by diabetes (22.1%). The prevalence of each of the selected chronic condition among study population is presented in Appendix A. Amongst the non-complex multimorbidity group obesity + oral conditions (7.9%) were the most commonly occurring dyad. The detailed presentation of the most common conditions contributing to non-complex multimorbidity is presented in Appendix A. The most frequently occurring conditions grouped by ICD-10 chapters were endocrine/nutritional/metabolic system + circulatory system + digestive system (13.3%). The detailed presentation of commonly occurring patterns of chronic conditions grouped by ICD-10 chapters contributing to complex multimorbidity is presented in Appendix A. 

According to the results of the GLM model, participants aged 75 years had a higher risk of having complex multimorbidity (PRR: 1.60 (1.44, 1.78)) than their younger counterparts (Table 3). Females (PRR: 1.10 (1.01, 1.19)); urban residents (PRR: 1.13 (1.03, 1.24)); lesser years of schooling, i.e., up to primary school (PRR: 1.10 (1.04, 1.18)); currently not working (PRR: 1.60 (1.46, 1.76)); and the richest group (PRR: 1.24 (1.08, 1.41)) were significantly associated with complex multimorbidity. Regarding age, the prevalence difference from the reference category 45–49 years for complex multimorbidity varied from (PRD: 14 (11, 17)) percentage points (pp) for participants aged 60–74 years to (PRD: 16 (12, 20)) pp for respondents aged ≥75 years. The highest difference of PRD from the reference category of no formal education was observed among participants who were educated up to primary school (PRD: 35 (1, 6)) pp and graduate and above (PRD: −5 (−12, 0.8)) pp. 

In the adjusted model, participants aged ≥75 years (APRR: 1.44 (1.30, 1.60)); urban residents (APRR: 1.10 (1.02, 1.20)); participants with education up to primary school (APRR: 1.12 (1.04, 1.20)); currently not working (APRR: 1.36 (1.24, 1.48)); and the richest group (APRR: 1.27 (1.12, 1.44)) were significantly associated with complex multimorbidity. After adjusting for various socio-demographic characteristics, we observed that the prevalence difference varied the highest among participants with education up to primary school (APRD: 4 (1, 6)) pp. We presented another fully adjusted model with changed references as Appendix A. Additionally, we also presented a comparison of a prevalence risk ratio with a prevalence odds ratio to justify the use of the former (represents no overestimation) in Appendix A.

Participants with complex multimorbidity had a significantly higher rate (*p* value < 0.001) of inpatient hospitalization (13.75 percent) than respondents without complex multimorbidity (10.16 percent). However, the number of outpatient visits reported by both groups was comparable (Table 4a). The mean number (mean ± sd) of inpatient visits among the complex multimorbidity group was 0.20 ± 0.67 as compared to 0.13 ± 0.49 among the non-complex multimorbidity group (Appendix A). The multivariable regression model adjusted for various socio-demographic correlates showed a higher chance of having an inpatient visit among complex multimorbidity [AOR: 1.30 (1.13–1.50)] as compared to those who have non-complex multimorbidity (Appendix A). Furthermore, patients with complex multimorbidity paid ₹ 800 for both outpatient services, compared to ₹ 600 for patients without complex multimorbidity (Table 4b). Similarly, respondents with complex multimorbidity had to pay approximately 10% more out of pocket (₹ 11,050) for inpatient hospitalizations than patients with non-complex multimorbidity (₹ 10,000). The healthcare expenditure converted from ₹ to United States Dollar (USD) is presented in the Appendix A. The healthcare expenditure on complex multimorbidity and non-complex multimorbidity as a proportion of India’s total healthcare expenditure Gross Domestic Product (GDP) is presented in the Appendix A.

The SRH of participants with non-complex multimorbidity was higher than that of participants with complex multimorbidity (Table 5). A greater proportion of non-complex multimorbid participants (3.16 percent) rated their health as excellent, compared to only 1.7 percent of complex multimorbid participants, whereas the latter reported a lower SRH by rating their health as poor (24.03 percent) compared to the formers (11.23 percent).

## 4. Discussion

### 4.1. Principal Findings

In contrast to multimorbidity, which considers the number of individual chronic conditions, complex multimorbidity considers multiple body systems which align with the recent recommendations of considering clusters of diseases [21,22]. Chronic diseases were found to be most prevalent in the digestive and circulatory systems in the study population. Surprisingly, one third of those who were multimorbid had complex multimorbidity. Participants over the age of 75, city dwellers, respondents with fewer years of schooling, the unemployed, and the affluent group were found to be significantly associated with complex multimorbidity. Healthcare utilisation and expenditure were higher, as expected, among respondents with complex multimorbidity, whereas SRH was lower when compared to non-complex multimorbid individuals. We discovered that one third of the multimorbid people had complex multimorbidity. Despite the term multimorbidity being discussed by Harrison et al. in 2014, [8] almost eight years ago, this is the first report from India on complex multimorbidity.

Our study considered complex multimorbidity among multimorbid individuals but most of the reports are based on population level estimates which made comparing our findings with similar studies challenging. The variation in multimorbidity pattern is determined not only by biological factors, but also by their interaction with social determinants [23]. In accordance with the World Health Organization’s (WHO) Commission on Social Determinants of Health (CSDH) framework [24], we discovered that complex multimorbidity was influenced by social determinants of health such as socioeconomic status, which includes social group, low educational attainment, and employment status (never worked/currently not working). Furthermore, intermediary determinants such as an individual’s behavioural and psychosocial factors play a role because compliance with healthy practices is dependent on these factors. Socioeconomic disparities, as well as an overburdened healthcare system, may stymie India’s efforts to achieve universal health coverage (UHC) [25]. Our findings show, however, that affluent people (the richest group of wealth quintile) are more likely to have complex multimorbidity, which could be attributed to their ability to obtain better diagnostic facilities through their ability to pay to see a medical doctor.

Although OPD visits were nearly equal, the complex multimorbidity group had significantly more inpatient hospitalizations than the non-complex multimorbidity group. A probable reason for nearly equal OPD visits could be that both multimorbid as well as complex multimorbid individuals receive timely routine care but the latter group requires higher inpatient services due to the involvement of multiple organ systems. In this case, the disparity in healthcare utilisation between the two groups may aid us in identifying individuals with greater healthcare needs, particularly in secondary care. Furthermore, categorizing diseases based on body systems has an advantage over the traditional system of disease entities or number of chronic conditions in that the former may provide information on the types of specialists involved or required for treatment. Although treating chronic conditions that affect the same body system is more complementary, dealing with multiple body systems may necessitate more coordinated care. Furthermore, we found that patients with complex multimorbidity had higher out-of-pocket expenditure (OOPE) as compared to the non-complex multimorbid group, which is consistent with the findings of a recent study that compared healthcare expenditure among participants aged 65–90 years across 11 high-income countries, revealing that expenditure increases incrementally among individuals with complex multimorbidity [26].

Furthermore, we found that patients with complex multimorbidity had a lower SRH than multimorbid individuals, which is consistent with the findings of a Norwegian study based on Helseundersokelsen i Nord-Trondelag (HUNT) data, which discovered that patients with complex multimorbidity required more assistance in instrumental activities of daily living [27]. This could be managed by providing continuity of care through integrated case management interventions, as demonstrated by the findings of a previous study in which case management interventions for patients with complex multimorbidity resulted in a significant decrease in emergency and unplanned visits [28].

### 4.2. Implications for Policy and Practice

This study gave evidence that complex multimorbidity significantly prevails among multimorbid individuals. Further, unplanned hospital admissions and other emergency service requirements of this group need to be assessed for allocating resources. Existing healthcare programmes such as the National Programme for Prevention and Control of Cancer, Diabetes, Cardiovascular Diseases and Stroke (NPCDCS) and the National Program for Health Care of the Elderly should envisage to identify patients with complex multimorbidity so as to provide timely and quality care to them. Primary care facilities should be strengthened to increase availability and accessibility while making the facilities affordable. Here, the newly established Health and Wellness Centres (HWCs) can be an opportunity where these identified individuals can be provided quality care. Additionally, these identified individuals can be registered with the use of an electronic platform at HWCs. Through this platform, they can easily be followed with their past medical records available anytime in case of an emergency. Disease specific guidelines should be revisited to introduce patient-centred care with a focus to reduce polypharmacy and increase quality of life among these patients.

### 4.3. Strengths and Limitations

We used nationally representative data to compare the patient outcomes of non-complex multimorbidity with complex multimorbidity which helped us in generating evidence on complex multimorbidity as a useful measure in identifying patients requiring additional attention and care. This study is the first attempt to identify a mutually exclusive group of complex multimorbid patients among those having multimorbidity. Furthermore, we classified diseases based on ICD-10 chapters, a widely accepted classification system. However, this study was limited by self-reported conditions which might lead to an underestimation of the true prevalence. Additionally, the participants might have not undergone proper diagnosis of all conditions owing to their socio-economic conditions which could further undermine the true prevalence of multimorbidity in this age group. We used self-rated health as a proxy indicator of health-related quality of life, but SRH can be confounded by socio-demographic and economic status of the participants which is another limitation of the study. We did not investigate behavioural factors such as use of tobacco, and physical activity. This study is based on the first round of longitudinal LASI survey (cross-sectional data); hence, it could not establish causality.

## 5. Conclusions

This study adapted the concept of complex multimorbidity to an Indian context and demonstrated that one third of older adults with multiple chronic conditions have complex multimorbidity. We found that social gradients have a substantial impact on the distribution of complex multimorbidity. In addition, this study revealed that patient outcomes, such as healthcare utilisation and expenditure, are higher among complex multimorbid individuals compared to multimorbid patients, while self-rated health deteriorates; complex multimorbidity may be a more effective method for identifying patients requiring substantial healthcare resources. This necessitates a systematic assessment of the healthcare needs of this vulnerable population, as well as future research into care-seeking pathways and prioritisation among this population, which can aid in policy-level resource allocation.

## Figures and Tables

**Figure 1 ijerph-19-09091-f001:**
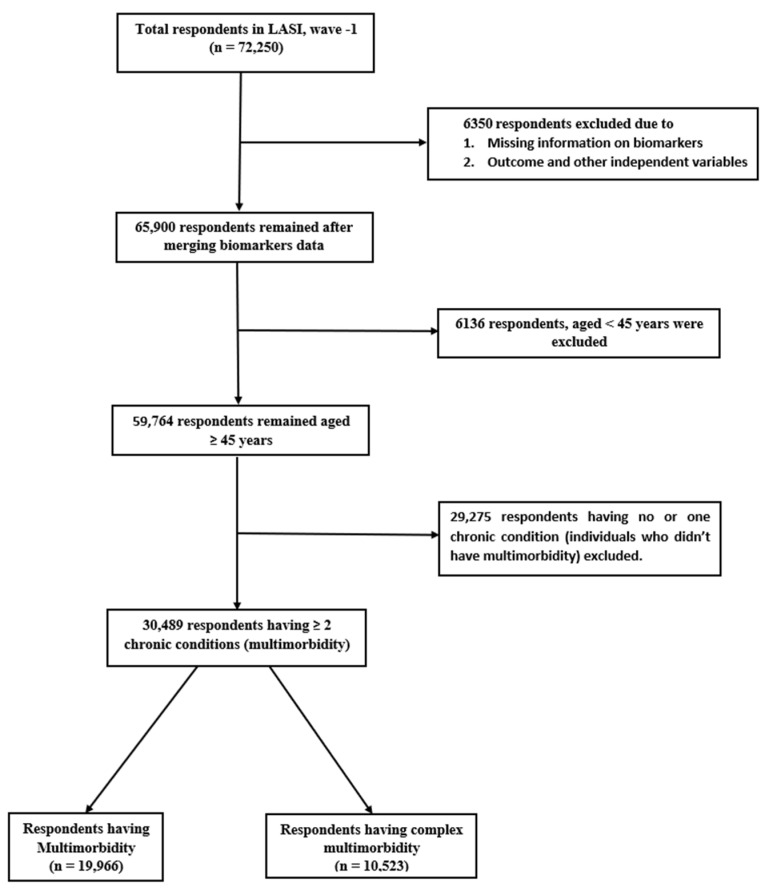
Selection of study population.

**Figure 2 ijerph-19-09091-f002:**
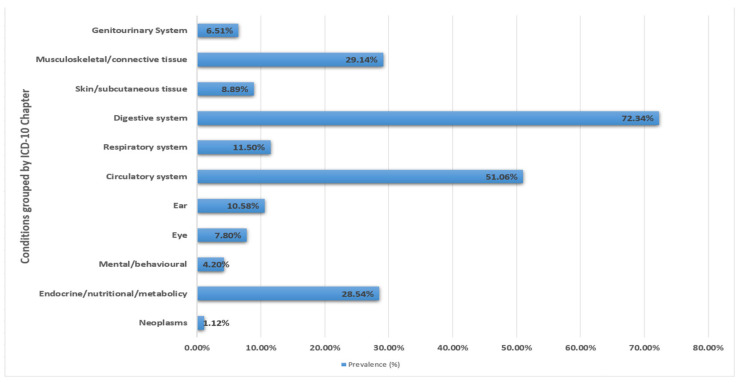
Prevalence of chronic conditions grouped by ICD-10 chapters.

**Table 1 ijerph-19-09091-t001:** Conditions grouped by ICD-10 chapter.

ICD-10 Chapter	Conditions Grouped
II Neoplasms	Cancer or malignant tumour
IV Endocrine/nutritional/metabolic	Hypercholesterolemia; Diabetes; thyroid disorders
V Mental/behavioural	Any neurological, or psychiatric problems such as depression, Alzheimer’s/Dementia, unipolar/bipolar disorders, convulsions, Parkinson’s
VII Eye	Cataract; glaucoma
VIII Ear/mastoid	Hearing impairment
IX Circulatory system	Hypertension; stroke; chronic heart disease (heart attack, myocardial infarction, congestive heart failure, or other chronic heart problems)
X Respiratory system	Chronic lung diseases such as asthma, chronic obstructive pulmonary disease/chronic bronchitis, or other chronic lung problems
XI Digestive system	Gastrointestinal problems (gastro esophageal reflux disease, constipation, indigestion, piles, peptic ulcer); oral Conditions (bleeding gums, swelling gums, loose teeth, dental caries)
XII Skin/subcutaneous tissue	Skin diseases
XIII Musculoskeletal/connective tissue	Arthritis or rheumatism, osteoporosis, or other bone/joint diseases
XIV Genitourinary system	Chronic kidney failure; incontinence

**Table 2 ijerph-19-09091-t002:** Socio-demographic characteristics of study population (unweighted distribution).

Socio-Demographic Characteristics		Total Population *n* (%)	Complex Multimorbidity *n* (%)	Non-Complex Multimorbidity *n* (%)
Age (*n* = 30,489)	Mean (±SD)	60.8 (±10.5)	63.25 (±10.5)	59.58 (±10.3)
	45–59 years	14,378 (47.2)	3883 (36.9)	10,495 (52.6)
	60–74 years	12,622 (41.4)	5037 (47.9)	7585 (38.0)
	75 years or more	3489 (11.4)	1603 (15.2)	1886 (19.4)
Gender (*n* = 30,489)	Male	12,986 (42.6)	4257 (40.4)	8729 (43.7)
	Female	17,503 (57.4)	6266 (59.6)	11,237 (56.3)
Residence (*n* = 30,489)	Rural	17,865 (58.6)	5857 (55.7)	12,008 (60.1)
	Urban	12,624 (41.4)	4666 (44.3)	7958 (39.9)
Social Group (*n* = 30,242)	Scheduled Tribes	4874 (16.1)	1034 (9.9)	2793 (14.1)
	Scheduled Castes	3827 (12.6)	1645 (15.7)	3229 (16.3)
	Other Backward Class	11,847 (39.2)	4217 (40.4)	7630 (38.5)
	Other Castes	9694 (32.1)	3546 (34.0)	6148 (31.1)
Education (*n* = 30,489)	No Formal Education	12,964 (42.5)	4414 (41.9)	8550 (42.8)
	Up to Primary	8050 (26.4)	2955 (28.1)	5095 (25.5)
	Middle school to Higher Secondary & Diploma	7717 (25.3)	2620 (24.9)	5097 (25.5)
	Graduation & Above	1758 (5.8)	534 (5.1)	1224 (16.2)
Employment status (*n* = 30,489)	Never Worked	9745 (32.0)	3678 (35.0)	6067 (30.4)
	Currently not working	9393 (30.8)	3966 (37.7)	5427 (27.2)
	Currently working	11,351 (37.2)	2879 (27.3)	8472 (42.4)
Wealth Index (*n* = 30,489)	Poorest	4786 (15.7)	1456 (13.8)	3330 (16.7)
	Poorer	5674 (18.6)	1847 (17.6)	3827 (19.2)
	Middle class	6070 (19.9)	2075 (19.7)	3995 (20.0)
	Richer	6720 (22.1)	2409 (22.9)	4311 (21.6)
	Richest	7239 (23.7)	2736 (26.0)	4503 (22.5)

**Table 3 ijerph-19-09091-t003:** Prevalence Risk Ratio (PRR) and Prevalence Risk Differences (PRD) and Adjusted Prevalence Risk Ratio (APRR) and Adjusted Prevalence Risk Differences (APRD) of complex multimorbidity across various socio-demographic characteristics.

Correlates		Complex Multimorbidity Present			
		PRR	PRD	APRR	APRD
Age	45–59 years	Reference			
	60–74 years	1.51 (1.37, 1.65)	0.14 (0.11, 0.17)	1.42 (1.29, 1.56)	0.11 (0.08, 0.14)
	75 years or more	1.60 (1.44, 1.78)	0.16 (0.12, 0.20)	1.44 (1.30, 1.60)	0.12 (0.08, 0.16)
Gender	Male	Reference			
	Female	1.10 (1.01, 1.19)	0.03 (0.005, 0.06)	1.05 (0.98, 1.14)	0.02 (−0.006, 0.04)
Residence	Rural	Reference			
	Urban	1.13 (1.03, 1.24)	0.04 (0.01, 0.08)	1.10 (1.02, 1.20)	0.04 (0.006, 0.07)
Social Group	Scheduled Tribes	Reference			
	Scheduled Castes	1.35 (1.18, 1.55)	0.09 (0.05, 0.13)	1.33 (1.16, 1.52)	0.07 (0.04, 0.11)
	Other Backward Class	1.37 (1.19, 1.59)	0.09 (0.54, 0.13)	1.27 (1.11, 1.46)	0.06 (0.02, 0.09)
	Other Castes	1.47 (1.30, 1.67)	0.12 (0.85, 0.15)	1.31 (1.14, 1.50)	0.07 (0.03, 0.11)
Education	No Formal Education	Reference			
	Up to Primary	1.10 (1.04, 1.18)	0.35 (0.01, 0.06)	1.12 (1.04, 1.20)	0.04 (0.01, 0.06)
	Middle school to Higher Secondary & Diploma	1.07 (1.94, 1.23)	0.25 (−0.02, 0.07)	1.09 (0.97, 1.24)	0.02 (−0.016. 0.06)
	Graduation & Above	0.84 (0.67, 1.04)	−0.05 (−0.12, 0.008)	0.86 (0.70, 1.05)	−0.06 (−0.12, 0.004)
Employment status	Never Worked	1.48 (1.31, 1.67)	0.12 (0.08, 0.16)	1.30 (1.17, 1.45)	0.07 (0.04, 0.11)
	Currently not working	1.60 (1.46, 1.76)	0.16 (0.13, 0.18)	1.36 (1.24, 1.48)	0.10 (0.07, 0.13)
	Currently working	Reference			
Wealth Index	Poorest	Reference			
	Poorer	1.04 (0.95, 1.14)	0.01 (−0.02, 0.04)	1.04 (0.95, 1.14)	0.01 (−0.01, 0.04)
	Middle class	1.05 (0.95, 1.16)	0.02 (−0.02, 0.05)	1.06 (0.96, 1.17)	0.02 (−0.01, 0.05)
	Richer	1.19 (1.07, 1.34)	0.06 (0.02, 0.10)	1.20 (1.08, 1.34)	0.06 (0.03, 0.10)
	Richest	1.24 (1.08, 1.41)	0.07 (0.02, 0.12)	1.27 (1.12, 1.44)	0.08 (0.04, 0.14)

**Table 4 ijerph-19-09091-t004:** (**a**) Association of selected indicators of healthcare utilisation with multimorbidity vs. complex multimorbidity among study population. (**b**) Healthcare expenditure in ₹ on multimorbidity vs. complex multimorbidity.

**(a)**				
**Indicators of Healthcare Utilisation**		**Non-Complex Multimorbidity** ***n*, % (95% CI)**	**Complex Multimorbidity** ***n*, % (95% CI)**	***p*-Value** **(Chi Square Test)**
Number of inpatient hospitalizations	None	13,462, 89.84 (89.34–90.32)	7839, 86.25 (85.53–86.96)	<0.001
	At least one	1522, 10.16 (9.68–10.65)	1249, 13.75 (13.04–14.46)	
Number of outpatient visits	None	2486, 16.77 (16.18–17.39)	1455, 16.28 (15.52–17.01)	0.32
	At least one	12,333, 83.23 (82.61–83.82)	7483, 83.72 (82.94–84.48)	
**(b)**				
**Type of Service Availed**		**Non-Complex Multimorbidity** **Median (IQR)**	**Complex Multimorbidity** **Median (IQR)**	
Outpatient visits		600 (260–1500)	800 (320–1850)	
Inpatient hospitalization		10,000 (3000–30,000)	11,050 (3470–32,250)	

**Table 5 ijerph-19-09091-t005:** Self-rated health of participants with multimorbidity vs. complex multimorbidity.

Self-Rated Health	Non-Complex Multimorbidity % (95%CI), *n*	Complex Multimorbidity % (95%CI), *n*	*p*-Value (Chi Square Test)
Excellent	3.16 (2.92–3.41), 626	1.7 (1.5–0.2), 182	<0.001
Very Good	15.8 (15.29–16.31), 3131	8.85 (8.32–9.4), 943	
Good	36.82 (36.15–37.50), 7299	26.71 (25.87–27.56), 2846	
Fair	33 (32.34–33.65), 6541	38.71 (37.78–39.64), 4125	
Poor	11.23 (10.80–11.70), 2227	24.03 (23.21–24.84), 2560	

## Data Availability

The dataset analysed during the current study is available in the LASI data repository held at ICT, IIPS [https://g2aging.org/?section=survey&surveyid=68] (accessed on 15 December 2021).

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
