# Peer review of "Multimorbidity and Complex Multimorbidity in India: Findings from the 2017–2018 Longitudinal Ageing Study in India (LASI)"

_ijerph, 2022, doi:10.3390/ijerph19159091_

Round 1
Reviewer 1 Report
The authors have studied complex multimorbidity from 30,489 multimorbid individuals from Longitudinal Ageing Study in India (LASI). They have defined complex multimorbidity as having multiple chronic conditions in different body organs. Interestingly, they found that most older multimorbid subjects have complex multimorbidity. Their finding also pointed out that subject’s socioeconomic circumstances and education have a role in complex multimorbidity. The data suggest that complex multimorbid patients use more health care resources and carry higher health costs. Their study greatly applies in policy making and resource allocation as it provides details to stratify older patients based on complex multimorbidity.
Specific concerns:
- The authors performed this study that answered the research question.
- Overall, this paper needs minor editing of grammar and language before being published.
- I wonder if the complex multimorbidity data is based on patient-reported chronic conditions. Given their socioeconomic situation, patients might not undergo proper diagnosis to reveal the actual number of chronic diseases that existed in the older patients.
- It could have been better to show a graphical representation of any correlation between aging patients and complex multimorbidity.
- In table1, it would be better to provide the number of patients in each category based on age, gender, etc.
Author Response
Reviewer 1
The authors have studied complex multimorbidity from 30,489 multimorbid individuals from Longitudinal Ageing Study in India (LASI). They have defined complex multimorbidity as having multiple chronic conditions in different body organs. Interestingly, they found that older multimorbid subjects have complex multimorbidity. Their finding also pointed out that subject’s socioeconomic circumstances and education have a role in complex multimorbidity. The data suggest that complex multimorbid patients use more health care resources and carry higher health costs. Their study greatly applies in policy making and resource allocation as it provides details to stratify older patients based on complex multimorbidity.
Specific concerns:
- The authors performed this study that answered the research question.
Author’s response: Thank you for your kind comments.
- Overall, this paper needs minor editing of grammar and language before being published.
Author’s response: Thank you for your suggestion. We have done all required edits with changes in grammar and punctuations.
- I wonder if the complex multimorbidity data is based on patient-reported chronic conditions. Given their socioeconomic situation, patients might not undergo proper diagnosis to reveal the actual number of chronic diseases that existed in the older patients.
Author’s response: This study is based on self-reported chronic conditions which we have already mentioned in the limitations section. However, we agree with the reviewer that patients might not have undergone proper diagnosis due to their socio-economic situation which we have now added as an additional limitation.
- It could have been better to show a graphical representation of any correlation between aging patients and complex multimorbidity.
Author’s response: We appreciate the reviewer’s suggestion. However, this is well-established in literature that complex multimorbidity and multimorbidity increases with age. Hence, we feel it will not be novel to show this correlation. However, we thank you again for thoughtful comment.
- In table1, it would be better to provide the number of patients in each category based on age, gender, etc.
Author’s response: Supplementary Table S3 depicts the socio-demographic characteristics of study population. However, as suggested by the reviewer we have now added it in main text as Table 2.
We thank the reviewer for thoughtful comments and valuable time.
Reviewer 2 Report
Dear Editor, Thanks for giving me the opportunity to revise this paper. In this manuscript, the authors explored prevalence factors associated with complex multimorbidity in India,where burden of multimorbidity is constantly raising. This study, using data from the Longitudinal Ageing Study in India (LASI), is the first addressing this topic in Indian individuals. Complex multimorbidity is the presence of at least 3 diseases involving 2 or more body systems and has shown to severely impact health care use, hospitalization risk and individual outcomes. Data from this study are quite interesting and gives a first picture on the prevalence and associated risks (increased hospitalization rates and costs) among Indian individuals of 45 years or more. The manuscript is well written in all its parts, I have only a minor concern to be addressed:
1) as multimorbidity and aging are closely related, I am wondering whether and how much data results may be affected by individuals' age. I suggest to briefly explore hierarchical clusters and factors associated with complex multimorbidity in distinct age groups (after stratifying by age category).
2) Minor changes:
-add "or more" to the last category of age groups in the method section ("75 or more").
-change ; with , in the line "While co-existence of two or more conditions was considered as non-complex multimorbidity; complex multimorbidity was defined as three or more chronic conditions among more than two body systems".
Author Response
Reviewer 2
Dear Editor, Thanks for giving me the opportunity to revise this paper. In this manuscript, the authors explored prevalence factors associated with complex multimorbidity in India, where burden of multimorbidity is constantly raising. This study, using data from the Longitudinal Ageing Study in India (LASI), is the first addressing this topic in Indian individuals. Complex multimorbidity is the presence of at least 3 diseases involving 2 or more body systems and has shown to severely impact health care use, hospitalization risk and individual outcomes. Data from this study are quite interesting and gives a first picture on the prevalence and associated risks (increased hospitalization rates and costs) among Indian individuals of 45 years or more. The manuscript is well written in all its parts, I have only a minor concern to be addressed:
- as multimorbidity and aging are closely related, I am wondering whether and how much data results may be affected by individuals' age. I suggest to briefly explore hierarchical clusters and factors associated with complex multimorbidity in distinct age groups (after stratifying by age category).
Author’s response: Thank you so much for your valuable input. However, the other two reviewers want hierarchical clusters be removed from the manuscript as it is not in line with the research question and outcomes of the study. We appreciate your valuable suggestion but also think in similar lines with the other reviewers and have removed cluster analysis from manuscript.
2) Minor changes:
add "or more" to the last category of age groups in the method section ("75 or more").
Author’s response: Thank you for pointing out this mistake. We have now changed it as suggested by the reviewer.
-change ; with , in the line "While co-existence of two or more conditions was considered as non-complex multimorbidity; complex multimorbidity was defined as three or more chronic conditions among more than two body systems".
Author’s response: Thank you for your suggestion. We have changed as suggested by the reviewer.
We thank reviewer for thoughtful comments and valuable time.
Reviewer 3 Report
This study used data collected in India in 2017-2018 to assess multimorbidity outcomes, distinguishing multimorbidity as complex vs non-complex based on the anatomical diversity of the conditions studied. Insufficient information is provided to fully evaluate the study and there are some concerns about the methods used and whether they were appropriate or validated for the study context. The major concern is that the definitions used for complex vs non-complex multimorbidity are not distinct and must be revised with the analysis rerun. I believe the authors have the data to do so and, with substantial revision, this analysis could make a useful contribution to the multimorbidity literature.
ABSTRACT
Update abstract to convey the exact definition of complex vs non-complex multimorbidity.
OUTCOME
Please justify the age cutoff of 45.
Provide additional information on the self-reported chronic conditions. How was the survey for these conditions validated? Did you use questions from the WHO STEPS questionnaire for NCDs? Were these self-diagnoses or diagnoses given by health workers?
The distinction between complex and non-complex multimorbidity is too blurry here. Either more details are needed, or the definitions need to be redone. For non-complex, are you restricting the 2+ conditions to the same system? If not, non-complex can overlap with complex multimorbidity. Why is there an arbitrary increase in the number of conditions for complex multimorbidity (why 3 instead of 2)? It seems that complex multimorbidity would be better off with the same number of conditions as non-complex (2+) but restricted to counting only conditions representing unique bodily systems.
The number of systems represented should be noted in the outcome. Table S2 should be referred to in the outcome section or better yet should be brought into the main text.
Justify also how/why the 17 conditions were chosen. If from a greater list of stated conditions, then give the breakdown from the original list to this more narrow selection.
COVARIATES
- Move all the definitions of sociodemographic variables to their own section.
STATISTICAL ANALYSIS
What was the point of the cluster analysis? It doesn’t feed into the definition of your outcome, which would have been a nice way to look at groupings of morbidities not reliant on ICD10 codes. I suggest removing this part of the paper as it doesn’t add anything.
Clarify what was done with chi2 tests; what constitutes a sociodemographic group is unclear and the tabulations done are not obvious.
Compare the prevalence ratio (PR) with the prevalence odds ratio (POR) – just to see if there is overestimation in the POR justifying the use of PR.
Justify the choice of the reference category for PRs given that the reference category can affect the point estimates; show in the supplement the differences in estimates with difference reference categories. The reference category of the outcome isn’t obvious.
What is meant by applying covariates separately? Are these univariate models? Was this for variable selection? If so, where is the fully adjusted multivariate model? You have enough observations to run a multivariate model and seem to have done so later on. Provide also the LASI survey weighting and design effects accounts for; the hierarchical nature of the data has not been presented.
Is the model using a log-binomial link? Clarify please.
RESULTS
Is the result for participants in urban areas consistent when reference categories for PRs are changed? The adjusted result is borderline given the CIs. Given the difference in health care access is descriptive only; it should not be presented as a main result in the abstract.
The results section presents unadjusted results in the main text write up but the abstract presents the adjusted results. The latter should be presented in both. Also, there is a need to discuss residual confounding as there is a massive difference in the coefficient magnitude (e.g. for the urban variable) in the unadjusted vs adjusted PR.
Why not investigate a full model predicting the health care variables as outcomes ) inpatient hospitalizations and self-rated health only since the other vars. have minor differences or none) and using multimorbidity as the key exposure adjusted for the sociodemographic characteristics?
Where is the tool used for self-rated health? Provide the questions in the supplement, indicate whether they were validated beforehand, and whether an international standardized survey was used for this (such as EQ5L etc). Self-rated health needs to be presented with extreme caution given it is highly confounded, usually, by sociodemographics/socioeconomic status.
Author Response
Reviewer 3
This study used data collected in India in 2017-2018 to assess multimorbidity outcomes, distinguishing multimorbidity as complex vs non-complex based on the anatomical diversity of the conditions studied. Insufficient information is provided to fully evaluate the study and there are some concerns about the methods used and whether they were appropriate or validated for the study context. The major concern is that the definitions used for complex vs non-complex multimorbidity are not distinct and must be revised with the analysis rerun. I believe the authors have the data to do so and, with substantial revision, this analysis could make a useful contribution to the multimorbidity literature.
ABSTRACT
Update abstract to convey the exact definition of complex vs non-complex multimorbidity.
Author’s response: Thank you for your suggestion. We have now incorporated the definition of complex vs non-complex multimorbidity in the abstract as suggested by the reviewer.
OUTCOME
Please justify the age cutoff of 45.
Author’s response: As suggested by the reviewer, we have now justified the age cutoff of 45 years under section 2.1 of the manuscript.
Provide additional information on the self-reported chronic conditions. How was the survey for these conditions validated? Did you use questions from the WHO STEPS questionnaire for NCDs? Were these self-diagnoses or diagnoses given by health workers?
Author’s response: Thank you for your suggestion. We have now revised methods section to incorporate more details on the methods employed by LASI survey to obtain the data.
The distinction between complex and non-complex multimorbidity is too blurry here. Either more details are needed, or the definitions need to be redone. For non-complex, are you restricting the 2+ conditions to the same system? If not, non-complex can overlap with complex multimorbidity. Why is there an arbitrary increase in the number of conditions for complex multimorbidity (why 3 instead of 2)? It seems that complex multimorbidity would be better off with the same number of conditions as non-complex (2+) but restricted to counting only conditions representing unique bodily systems.
Author’s response: We appreciate the comments of reviewer. In this study, we have used the conventional and widely accepted definition of multimorbidity i.e. two or more chronic conditions without defining the index disease. As suggested by the reviewer we have now described it more explicitly in the first line of the introduction for better understandability. This conventional definition has been used to select multimorbid participants as per the inclusion criteria of study as well as to define non-complex multimorbidity. For non-complex, the two or more chronic conditions can be across any body systems i.e. it is irrespective of body systems.
Additionally, in the absence of any universally accepted criteria to define complex multimorbidity, we defined it as three or more chronic conditions across more than two body systems as enumerated by Harrison C et al, 2014.
We further want to clarify that these two groups i.e. non-complex and complex are mutually exclusive which could be seen in the figure 1 (selection of study population).
We have now revised several lines in the manuscript to make non-complex and complex distinct for better readability and understandability.
The number of systems represented should be noted in the outcome. Table S2 should be referred to in the outcome section or better yet should be brought into the main text.
Author’s response: As suggested by the reviewer, we have now moved Table S2 to the main text.
Justify also how/why the 17 conditions were chosen. If from a greater list of stated conditions, then give the breakdown from the original list to this more narrow selection.
Author’s response: All 17 chronic conditions included by LASI were chosen to assess multimorbidity.
COVARIATES
- Move all the definitions of sociodemographic variables to their own section.
Author’s response: Thank you for your suggestion. However, we were not able to locate any socio-demographic variable apart from section 2.2. Kindly guide forward on specific variable which needs to be mentioned in that section.
STATISTICAL ANALYSIS
What was the point of the cluster analysis? It doesn’t feed into the definition of your outcome, which would have been a nice way to look at groupings of morbidities not reliant on ICD10 codes. I suggest removing this part of the paper as it doesn’t add anything.
Author’s response: Thank you for the suggestion. We have now removed it from results.
Clarify what was done with chi2 tests; what constitutes a sociodemographic group is unclear and the tabulations done are not obvious.
Author’s response: We have now mentioned this in detail in the main text under section 2.4.
Compare the prevalence ratio (PR) with the prevalence odds ratio (POR) – just to see if there is overestimation in the POR justifying the use of PR.
Author’s response: The reviewer has raised a valid point here and we did calculate the prevalence odds ratio (POR) at the stage of Statistical Analysis Planning (SAP). As the prevalence of outcome variable was >10%, we analysed for Prevalence Risk Ratio (PRR) that is more appropriate in such conditions. Also, the table for POR has been added this time in the supplementary file.
Justify the choice of the reference category for PRs given that the reference category can affect the point estimates; show in the supplement the differences in estimates with difference reference categories. The reference category of the outcome isn’t obvious.
Author’s response: We appreciate the points the reviewer has mentioned here. Primarily, we selected the reference categories based on the gradient of vulnerability. Like, the schedule tribes are more vulnerable than the scheduled caste; poorest among wealth quintile; no formal education; rural. Similarly, the females were found to be at more risk to develop several NCDs in earlier research hence, males were kept as reference. Participants with a higher age group are at a higher risk of having multimorbidity hence, lowest age group was kept as reference.
However, as suggested by the reviewer we have added a supplementary table after changing the refernces.
What is meant by applying covariates separately? Are these univariate models? Was this for variable selection? If so, where is the fully adjusted multivariate model? You have enough observations to run a multivariate model and seem to have done so later on.
Author’s response: Thank you for raising this concern. We have now changed “applying covariate separately” to form a better sentence. Yes, first we have seen the association of each covariate with complex multimorbidity separately i.e. univariate model presented as PRR and PRD. Followed by which we have analyzed a fully adjusted model presented as APRR and APRD.
Provide also the LASI survey weighting and design effects accounts for; the hierarchical nature of the data has not been presented.
Author’s response: This has now been added in the methods section along with a flow diagram as supplementary file.
Is the model using a log-binomial link? Clarify please.
Author’s response: We utilized “binreg” link in generalized linear models (GLM) to investigate the regression models.
RESULTS
Is the result for participants in urban areas consistent when reference categories for PRs are changed? The adjusted result is borderline given the CIs.
Author’s response: We did run the regression model changing the reference categories. The results remained consistent for the “residence” variable. We also have provided a table changing the references of other variables in the supplementary file.
Given the difference in health care access is descriptive only; it should not be presented as a main result in the abstract.
Author’s response: We agree with the reviewer that the healthcare access difference is descriptive, however we have also seen the statistical significance through chi-square test. This is an important finding of this manuscript which needs to be highlighted and hence, we plan to keep it in abstract part. However, we are open to further revisions if required.
The results section presents unadjusted results in the main text write up but the abstract presents the adjusted results. The latter should be presented in both.
Author’s response: Thank you for pointing out this mistake. Due to typing error ‘A’ for adjusted was not written which has now been corrected hence, adjusted results are mentioned in the main text.
Also, there is a need to discuss residual confounding as there is a massive difference in the coefficient magnitude (e.g. for the urban variable) in the unadjusted vs adjusted PR.
Author’s response: Thank you for raising this concern. However, we note that no such massive difference in the coefficient magnitude for the urban variable was encountered, between adjusted [PRR: 1.10 (1.02, 1.20)] and unadjusted models [APRR: 1.13 (1.03, 1.24)]. We still request the reviewer to provide further feedback, if we made any mistake in understanding this concern.
Why not investigate a full model predicting the health care variables as outcomes) inpatient hospitalizations and self-rated health only since the other vars. have minor differences or none) and using multimorbidity as the key exposure adjusted for the sociodemographic characteristics?
Author’s response: We have now investigated a full model using logistic regression.
Where is the tool used for self-rated health? Provide the questions in the supplement, indicate whether they were validated beforehand, and whether an international standardized survey was used for this (such as EQ5L etc). Self-rated health needs to be presented with extreme caution given it is highly confounded, usually, by sociodemographic/socioeconomic status.
Author’s response: Self-rated health is based on the participant’s perception of their health. However, self-rated health is widely used as a proxy indicator of health related quality of life and hence, we chose this variable. However, we agree with the concern of the reviewers and for this reason we have now mentioned this as one of the limitations of this study.
We thank the reviewer for valuable suggestions and time.
Reviewer 4 Report
The research work evaluates the prevalence and patient outcomes of complex multimorbidity, compared to non complex multimorbidity, among people aged 45 and more in India. Data from a panel study Longitudinal Ageing Study India (LASI) is used.
The topic is interesting and the manuscript is well written. However there are major concerns and critical points.
First of all, the outcome. Complex comorbidity is not properly described. The authors stated in introduction that identifying individuals with complex multimorbidity among multimorbid people is important to focus on individuals who need additional care and support. However the study did not highlight the baseline characteristics of complex multimorbid patients neither describe them. In fact Figure 1 and 2 referred to chronic conditions but not complex multimorbid. Figure 2 helps to identify clusters but then how clusters are related to the outcome definition? Therefore no results are shown to characterise the complexity of multimorbid patients. In supplementary Table S2 the aggregation to identify organ system is reported. So complex multimorbid patients should have at least 2 diseases in different groups. But which are the most common combinations?
Results reported in the conclusions "this study revealed that patient out-comes, such as healthcare utilization and expenditures, are higher among complex multi-morbid individuals compared to multimorbid patients" are quite obvious. Nobody would suppose otherwise. Therefore without a more appropriate characterization and description of complex comorbidity the findings of this study would not so much helpful.
In detail, my comments:
Abstract.
1) Please change APD with APRD, in line with the Table results.
2) Please avoid IPD term here because is not necessary to report in the abstract and also because otherwise should be better specified as "in patient department".
Introduction
1) Please better explain the sentence "Individual conditions take precedence over a patient-centered approach in India's healthcare system". Do you mean that a holistic approach is missing or that symptoms are most important than the person to be cared of? Please also give a better reference for that because in the article you quoted no patient centered approach is mentioned.
Materials and Methods
1) There is a reason for choosing to include respondents from 45 years in LASI? Why this cut-off age?
2) If in LASI only people older than 44 are considered, why in supplemental, Flowchart Figure S1 you need to exclude respondents < 45 years of age?
3) Please remove viz. not necessary.
4) Verbal data are mentioned (do you mean interviews?) Are data from interviews analysed? How were variable recorded?
5) Wealth categories should be defined. What do "2,3, 4" mean? Also in Supplemental Table S3 there is no description.
6) Please report here the abbreviation SHR (self reported health) because later in the results is reported without explanation.
7) How "excellent, very good, fair ..". were defined? Were only patient feelings or were based on quality of life questions? If they are only patient feelings I don't think that are good indicators without a more deepen qualitative analysis of outcomes and without knowing a whole patient story.
8) Please rephrase "We total the number of system-specific chapters in which a respondent had at least one chronic condition in any of the system-specific chapters." It is not clear.
9) How were blood sample and other laboratory test results analysed? Were they combined together with the self reported chronic conditions to define a disease (i.e. hypertension)? If not, as I guess, why was it not possible?
Statistical analysis
1) Please replace "is used" with "was used". Please use always the past in the text (check throughout all document).
2) Did you use a Poisson model with a log(offset) since it is a rate model? Please specify it. In this case, have you checked that Poisson assumptions were not violated (ie. no overdispersion)?
3) Please replace APD with APRD because in Tables you used APRD.
4) Please describe LASI survey weighting. It is not clear what they are.
Results
1) Please report the number and % of complex multimorbid over total sample size. It was reported in the abstract but not in the results.
2) Figure 2 please improve the quality of the Figure. Remove background color and change the labels avoiding "_". However Figure 2 like it is, it does not help much to identify the outcome, though it would help so much the visualization if related to the main outcome definition.
3) Please use comma or semicolon to separate values within brackets when reporting confidence intervals (please change also in the abstract) because "-" is confusing, since PRD and APRD are sometimes negative.
4) I suggest to report PRD and APRD in % also in Tables, in line with what reported in the main text.
5) Please specify that the prevalence difference was calculated respect to the reference category i.e. the sentence "The prevalence difference for complex multimorbidity varied from" should be "The prevalence difference, from each predictor reference category, for complex multimorbidity varied from".
6) Please don't report in the text the difference PRD and APRD of "graduate and above" respect to "no formal education" because is not significant (the confidence interval contains 0).
7) Looking at the tables, some results reported in text are wrong: urban residents [PRR: 1.35 (1.18-1.55)]; lesser years of schooling i.e. up to primary school [PRR: 1.35 (1.04-1.18)]. Please correct them. Moreover, replace 136 (1.24-1.48) with 1.36 (1.24-1.48)
8) Please report p <0.001 after "a significantly higher rate" when comparing number of inpatient hospitalizations. Moreover, quote Table 2.a after reporting this first result from it.
9) Please also report number of inpatient hospitalizations and outpatient visits mean and SD and make the two group comparisons. In fact it is not proper to describe them only as categorical variables.
10) Please specify INR (Indian Rupie). Please also report values converted in USD.
11) Why do not have you tested median (iqr) differences among the two groups? Moreover, could the authors give an idea on how much is the cost burden in terms of total health expenditure GDP? Looking only at the amount of money, is not so much informative.
12) Baseline characteristic comparisons among the two multimorbid group (complex and not) should be also reported.
13) Table 3. Please report, for more readibility, results as % (95 CI), n. So that proportions are highlighted at first.
Discussion
1) "Chronic diseases were found to be most prevalent in the digestive and circulatory systems in the study population". This result, as reported, does not help to understand the complexity, unless the authors mean that the most common complex comorbid pattern was digestive and circulatory system. In this case, the authors should better clarify it and overall the text and maybe this would also address my main concerns about outcome definition.
2) The reader has no idea about affluent group. Definition is missing (see previous comment above).
3) The authors compare their findings with the Brasilian study without mentioning that the Brasilian study also included people < 45 years and regarded only rural workers. Therefore, I doubt that results could be directly compared. Moreover, the results compared regard only the frequency of chronic conditions without referring to the complex pattern comorbidity. Moreover, please discuss the differences between the findings of this study and the findings of high-income countries studies.
4) Authors wrote "intermediary determinants such as an individual's behavioral and psychosocial factors play a role because compliance with healthy practices is dependent on these factors". However in this study no individual behavioral and psychosocial factors are investigated. Therefore the authors should report this in the "Strength and limitations" subsection.
5) Do the authors have a possible explanation for the result that OPD were not significantly different among the two groups?
6) "higher out of pocket". Please specify compared to non complex multimorbid.
7) What HUNT does it stand for in their study? Please specify it or remove it.
8) Please replace "generated evidence" with "gave evidence".
9) Among the limitations, also exclusion of people below 45 years of age should be mentioned, unless the main focus of this study was to focus on this category but, of course, limiting the interpretability of the findings.
Author Response
Reviewer 4
The research work evaluates the prevalence and patient outcomes of complex multimorbidity, compared to non complex multimorbidity, among people aged 45 and more in India. Data from a panel study Longitudinal Ageing Study India (LASI) is used.
The topic is interesting and the manuscript is well written. However there are major concerns and critical points.
First of all, the outcome. Complex comorbidity is not properly described. The authors stated in introduction that identifying individuals with complex multimorbidity among multimorbid people is important to focus on individuals who need additional care and support. However the study did not highlight the baseline characteristics of complex multimorbid patients neither describe them. In fact Figure 1 and 2 referred to chronic conditions but not complex multimorbid. Figure 2 helps to identify clusters but then how clusters are related to the outcome definition? Therefore no results are shown to characterise the complexity of multimorbid patients. In supplementary Table S2 the aggregation to identify organ system is reported. So complex multimorbid patients should have at least 2 diseases in different groups. But which are the most common combinations?
Author’s response: Thank you for reviewing our manuscript and providing valuable inputs to improve it. We would like to clarify that, in the absence of any universally accepted criteria to define complex multimorbidity, we defined it as three or more chronic conditions across more than two body systems as enumerated by Harrison C et al, 2014. We further want to clarify that the two groups i.e. non-complex and complex are mutually exclusive which could be seen in the figure 1 (selection of study population). We have now revised several lines in the manuscript to make non-complex and complex distinct for better readability and understandability. Additionally, as suggested by the reviewer we have added a new table (Table 2) depicting the baseline characteristics of both these groups.
Although cluster analysis showed the most common conditions grouped across body systems i.e. represented complex multimorbidity but as suggested by the reviewer that figure 2 (cluster analysis) is not related to the outcome, we have removed it from the manuscript.
Results reported in the conclusions "this study revealed that patient out-comes, such as healthcare utilization and expenditures, are higher among complex multi-morbid individuals compared to multimorbid patients" are quite obvious. Nobody would suppose otherwise. Therefore without a more appropriate characterization and description of complex comorbidity the findings of this study would not so much helpful.
Author’s response: Thank you for raising this concern. We have already mentioned this in the introduction section that the evidence from HICs suggest higher healthcare utilization and expenditure among complex multimorbid individuals. However, till date no study on complex multimorbidity has been done in India. The evidence on prevalence and correlates of complex multimorbidity will help in identifying the groups requiring higher and emergency health needs. Furthermore, establishing higher healthcare utilization and expenditure will help in drawing the attention of programme and policy makers towards the needs of complex multimorbid individuals.
In detail, my comments:
Abstract.
- Please change APD with APRD, in line with the Table results.
Author’s response: Thank you for pointing it out, we have done the needful.
2) Please avoid IPD term here because is not necessary to report in the abstract and also because otherwise should be better specified as "in patient department".
Author’s response: Thank you for the suggestion. We have specified it by enumerating it “in patient department”.
Introduction
1) Please better explain the sentence "Individual conditions take precedence over a patient-centered approach in India's healthcare system". Do you mean that a holistic approach is missing or that symptoms are most important than the person to be cared of? Please also give a better reference for that because in the article you quoted no patient centered approach is mentioned.
Author’s response: Thank you for your suggestion. We have changed this sentence for better clarity and understanding.
Materials and Methods
- There is a reason for choosing to include respondents from 45 years in LASI? Why this cut-off age?
Author’s response: In India, chronic conditions usually present around 45 years of age, a decade earlier than most of the high income countries, hence this age group was chosen. As suggested by the reviewer we have added this under section 2.1.
- If in LASI only people older than 44 are considered, why in supplemental, Flowchart Figure S1 you need to exclude respondents < 45 years of age?
Author’s response: LASI considered the individuals aged 45 years and above along with their spouses (irrespective of age). We have now mentioned this in text also for better clarity.
3) Please remove viz. not necessary.
Author’s response: We have removed “viz.” from the text.
- Verbal data are mentioned (do you mean interviews?) Are data from interviews analysed? How were variable recorded?
Author’s response: Data were collected through face to face participant interviews. We have now given a detail record of how each variable were collected in section 2.2 of the methods.
5) Wealth categories should be defined. What do "2,3, 4" mean? Also in Supplemental Table S3 there is no description.
Author’s response: We have now presented how wealth quintile was formed. Additionally, based on your suggestion, we have classified wealth index as Poorest, poorer, middle, richer and richest as provided in LASI India report.
6) Please report here the abbreviation SHR (self reported health) because later in the results is reported without explanation.
Author’s response: We have made the changes and have reported the abbreviation now.
7) How "excellent, very good, fair ..". were defined? Were only patient feelings or were based on quality of life questions? If they are only patient feelings I don't think that are good indicators without a more deepen qualitative analysis of outcomes and without knowing a whole patient story.
Author’s response: Self-rated health is based on the participant’s perception of their health. However, self-rated health is widely used as a proxy indicator of health related quality of life and hence, we chose this variable. However, we agree with the concern of the reviewers and for this reason we have now mentioned this as one of the limitations of this study.
8) Please rephrase "We total the number of system-specific chapters in which a respondent had at least one chronic condition in any of the system-specific chapters." It is not clear.
Author’s response: Thanks for your remark, we have reformed the sentence to bring out a clear picture.
9) How were blood sample and other laboratory test results analysed? Were they combined together with the self-reported chronic conditions to define a disease (i.e. hypertension)? If not, as I guess, why was it not possible?
Author’s response: Blood sample and other laboratory test results were not analysed in the study as these are not available in public domain. However, we used height and weight (anthropometric measures) from biomarker dataset file to calculate BMI.
Statistical analysis
1) Please replace "is used" with "was used". Please use always the past in the text (check throughout all document).
Author’s response: We have replaced the part as suggested.
2) Did you use a Poisson model with a log(offset) since it is a rate model? Please specify it. In this case, have you checked that Poisson assumptions were not violated (ie. no overdispersion)?
Author’s response: Here, we have not used the rate model. The model was run using link ‘binreg’ to calculate the prevalence risk ratio. We made rectified the typos replaced the rate with risk where ever applicable.
- Please replace APD with APRD because in Tables you used APRD.
Author’s response: Thank you for pointing this. We have replaced APD with APRD.
4) Please describe LASI survey weighting. It is not clear what they are.
Author’s response: We have described the complex nature of sampling adopted to give a nationally representative data by LASI in the section 2.1 (supplementary figure also). To compensate this, survey weights were taken into consideration during analysis.
Results
1)Please report the number and % of complex multimorbid over total sample size. It was reported in the abstract but not in the results.
Author’s response: We have mentioned the same in the last line of 2nd paragraph.
2) Figure 2 please improve the quality of the Figure. Remove background color and change the labels avoiding "_". However, Figure 2 like it is, it does not help much to identify the outcome, though it would help so much the visualization if related to the main outcome definition.
Author’s response: Thanks for pointing the issue. We are removing the figure as suggested.
3) Please use comma or semicolon to separate values within brackets when reporting confidence intervals (please change also in the abstract) because "-" is confusing, since PRD and APRD are sometimes negative.
Author’s response: We have made the changes as recommended.
4) I suggest to report PRD and APRD in % also in Tables, in line with what reported in the main text.
Author’s response: In tables, it is crude result whereas in main text it has been presented with an aim to interpret. We find this to have clarity and does not require a change.
5) Please specify that the prevalence difference was calculated respect to the reference category i.e. the sentence "The prevalence difference for complex multimorbidity varied from" should be "The prevalence difference, from each predictor reference category, for complex multimorbidity varied from".
Author’s response: Changed as suggested. Thank you
6) Please don't report in the text the difference PRD and APRD of "graduate and above" respect to "no formal education" because is not significant (the confidence interval contains 0).
Author’s response: We have removed that part from text.
7) Looking at the tables, some results reported in text are wrong: urban residents [PRR: 1.35 (1.18-1.55)]; lesser years of schooling i.e. up to primary school [PRR: 1.35 (1.04-1.18)]. Please correct them. Moreover, replace 136 (1.24-1.48) with 1.36 (1.24-1.48)
Author’s response: Thank you for mentioning. We have made the corrections.
8) Please report p <0.001 after "a significantly higher rate" when comparing number of inpatient hospitalizations. Moreover, quote Table 2.a after reporting this first result from it.
Author’s response: Thanks for pointing it. We have added it to the text. Table 2a has been quoted after the next line which marks the end of its description.
9) Please also report number of inpatient hospitalizations and outpatient visits mean and SD and make the two group comparisons. In fact it is not proper to describe them only as categorical variables.
Author’s response: Thank you for the suggestion. We have added a supplementary file representing the mean visits and SD.
10) Please specify INR (Indian Rupie). Please also report values converted in USD.
Author’s response: Thank you for the suggestion. We made the changes accordingly and presented it
11) Why do not have you tested median (iqr) differences among the two groups?
Author’s response: We have reported median and IQR for healthcare expenditure and have not tested its differences.
Moreover, could the authors give an idea on how much is the cost burden in terms of total health expenditure GDP? Looking only at the amount of money, is not so much informative.
Author’s response: We have presented it as a supplementary file.
12) Baseline characteristic comparisons among the two multimorbid group (complex and not) should be also reported.
Author’s response: As suggested we have provided a comparison of complex and non-complex multimorbid groups in the table.
13) Table 3. Please report, for more readibility, results as % (95 CI), n. So that proportions are highlighted at first.
Author’s response: Thank you for the suggestion, we have made the changes accordingly.
Discussion
- "Chronic diseases were found to be most prevalent in the digestive and circulatory systems in the study population". This result, as reported, does not help to understand the complexity, unless the authors mean that the most common complex comorbid pattern was digestive and circulatory system. In this case, the authors should better clarify it and overall the text and maybe this would also address my main concerns about outcome definition.
Author’s response: We do not intend to show complexity by reporting the prevalence of grouped chronic conditions on the basis of ICD-10 chapters, instead we want to highlight the similarity of results. However, as pointed by the reviewer regarding comparability of these studies due to age difference in sample size, we have removed it. We presented the patterns in dendrogram however,as per reviewer’s suggestion we have removed it.
2) The reader has no idea about affluent group. Definition is missing (see previous comment above).
Author’s response: We have now defined it. Thank you for pointing this out.
3) The authors compare their findings with the Brasilian study without mentioning that the Brasilian study also included people < 45 years and regarded only rural workers. Therefore, I doubt that results could be directly compared. Moreover, the results compared regard only the frequency of chronic conditions without referring to the complex pattern comorbidity. Moreover, please discuss the differences between the findings of this study and the findings of high-income countries studies.
Author’s response: Since, we investigated complex multimorbidity among multimorbid individuals it is a bit challenging to compare our findings with population based prevalence of complex multimorbidity which we have already mentioned in the discussion section.
4) Authors wrote "intermediary determinants such as an individual's behavioral and psychosocial factors play a role because compliance with healthy practices is dependent on these factors". However in this study no individual behavioral and psychosocial factors are investigated. Therefore the authors should report this in the "Strength and limitations" subsection.
Author’s response: Thank you for the suggestion. We have now added this as a limitation.
5) Do the authors have a possible explanation for the result that OPD were not significantly different among the two groups?
Author’s response: We have now mentioned the probable reason for nearly equal OPD visits.
6) "higher out of pocket". Please specify compared to non-complex multimorbid.
Author’s response: We have specified as suggested.
7) What HUNT does it stand for in their study? Please specify it or remove it.
Author’s response: We have enumerated its full form.
8) Please replace "generated evidence" with "gave evidence".
Author’s response: we have changed as suggested by the reviewer.
9) Among the limitations, also exclusion of people below 45 years of age should be mentioned, unless the main focus of this study was to focus on this category but, of course, limiting the interpretability of the findings.
Author’s response: Thank you for observing this. However, the data included is representative of only 45 years and above population whereas respondents below 45 years are not representative as the spouses of head of household were included irrespective of age i.e. <45 years also. Hence, this will not affect the interpretability of the findings and will not be a limitation.
We thank you again for valuable time and suggestions.
Round 2
Reviewer 3 Report
The authors have improved the manuscript. But, there is still major confusion as to the definitions of complex and non-complex multimorbidities. The convention is that multimorbidities is 2+ conditions. With the revision the authors introduce another definition of 3+ conditions. There is no justification to this. The population needs to be broken down by showing a flowchart of all individuals - then those with any conditions vs no conditions then among those with any conditions - those with any multimorbidity (2+ conditions) vs those without - then among any multimorbidity - those with complex (defined as 2+ anatomical systems) vs those with non-complex (2+ in same system only) [these are mutually exclusive].
Then the analysis should be rerun accordingly. Also, justification is needed for the 17 conditions studied - saying it was chosen by LASI is not enough. What is the scientific and medical basis?
If there is not already a table, the prevalence as well of each of the 17 conditions in the population must be provided as well as a breakdown of the most common conditions contributing to complex vs non-complex multimorbidity.
There also should be more literature cited about studies that look at multimorbidity or comorbidity within anatomical systems (this is common in the mental health disorders or for diabetes etc. - see editorials written by Chris Whitty in the Lancet or BMJ). The proposed definitions here are *not* unique and should be justified with existing literature.
Author Response
Reviewer 3
The authors have improved the manuscript. But, there is still major confusion as to the definitions of complex and non-complex multimorbidities. The convention is that multimorbidities is 2+ conditions. With the revision the authors introduce another definition of 3+ conditions. There is no justification to this. The population needs to be broken down by showing a flowchart of all individuals - then those with any conditions vs no conditions then among those with any conditions - those with any multimorbidity (2+ conditions) vs those without - then among any multimorbidity - those with complex (defined as 2+ anatomical systems) vs those with non-complex (2+ in same system only) [these are mutually exclusive].
Then the analysis should be rerun accordingly.
Author’s response: Thank you so much for this suggestion. The below depicted flowchart (Figure 1) is in accordance with the suggestions of reviewer. However, we have now understood the reviewer’s point of view w.r.t. the definition of complex multimorbidity and we apologize for not being able to clearly convey the definition.
Please refer to the following published papers:
Vinjerui KH, Bjerkeset O, Bjorngaard JH, Krokstad S, Douglas KA, Sund ER. Socioeconomic inequalities in the prevalence of complex multimorbidity in a Norwegian population: findings from the cross-sectional HUNT study. BMJ open. 2020 Jun 1;10(6):e036851.
Harrison C, Henderson J, Miller G, Britt H. The prevalence of complex multimorbidity in Australia. Australian and New Zealand journal of public health. 2016 Jun;40(3):239-44.
Our definition of non-complex as well as complex multimorbidity is in line with most of the international peer-reviewed published articles. However, in the section 2.3, we have now changed the operational definition so as to make the methods uniform in line with what has been done. Also, the same is reflected in the abstract too.
Operational definition
Complex Multimorbidity: We say at least three out of eleven system specific chapters are aggregated i.e. more than two body system. Each chapter/body system consists of one or more chronic condition (table 1), hence in three chapters (systems) at least three or more chronic conditions have come justifying our definition “complex multimorbidity [8], which is defined as the co-occurrence of three or more chronic conditions affecting more than two body systems in a single individual.”
Non-complex Multimorbidity: Now refer to Figure 1 where out of 30489 respondents having two or more chronic conditions we grouped 10,523 participants as complex based on above criteria. Then, 19,966 who had two or more chronic conditions across less than two body system (as all participants having chronic conditions across 3 or more body system have already been put in complex group leaving only participants with chronic conditions in one or two body system) have been referred as non-complex.
We hope that we are now able to do justice with the reviewer’s expectations. However, if required we are open to further revision.
Also, justification is needed for the 17 conditions studied - saying it was chosen by LASI is not enough. What is the scientific and medical basis?
Author’s response: Thank you for your suggestion. We have now added the reasons for taking these 17 conditions in the study. (We considered eighteen most commonly prevalent chronic conditions based on an extensive literature search (18).)
If there is not already a table, the prevalence as well of each of the 17 conditions in the population must be provided as well as a breakdown of the most common conditions contributing to complex vs. non-complex multimorbidity.
Author’s response: We have now added a supplementary file stating the prevalence of each of the 17 chronic conditions. Additionally, we have also added the breakdown of most common conditions (pattern analysis) contributing to non-complex and complex multimorbidity. (The most prevalent chronic condition was hypertension (47.9%) followed by diabetes (22.1%). The prevalence of each of the selected chronic condition among study population is presented in supplementary table S1. Amongst the non-complex multimorbidity group obesity + oral conditions (7.9%) were the most commonly occurring dyad. The detailed presentation of most common conditions contributing to non-complex multimorbidity is presented in supplementary table S2. The most frequently occurring conditions grouped by ICD-10 chapters were endocrine/nutritional/metabolic system + circulatory system + digestive system (13.3%). The detailed presentation of commonly occurring patterns of chronic conditions grouped by ICD-10 chapters contributing to complex multimorbidity is presented in supplementary table S3.)
There also should be more literature cited about studies that look at multimorbidity or comorbidity within anatomical systems (this is common in the mental health disorders or for diabetes etc. - see editorials written by Chris Whitty in the Lancet or BMJ). The proposed definitions here are *not* unique and should be justified with existing literature.
Author’s response: As suggested by the reviewer, we have cited few more papers which have explored multimorbidity within anatomical systems. (Reference No-20-23)
We thank you for your valuable inputs and time.
Reviewer 4 Report
The manuscript has improved, but there are still some important changes that should have been done and the authors did not fully answered my comments.
1. The outcome definition reported in the abstract " Complex multimorbidity refers to the co-occurrence of three or more chronic illnesses across >2many body systems" is not in line with the one the authors used later on section 2.3 "Individuals with complex multimorbidity were defined as those who scored at least three out of eleven system-specific chapters". So please clarify it and make some examples of complex comorbid patient.
2. The wealth index should be better specified. You should report the level of income (as intervals) that defined each category. Otherwise it is not clear what do you mean with Poorest, Poorer ...
3. In Table 2 The column Overall (for all the population) should be added otherwise it's not clear and also in text is not clearly reported i.e. "The majority of the participants came from rural areas, had no formal education, and were currently employed" we cannot see it from Table 2 because the two groups presented different results.
Please also change the format of Table 2, to be in line with the other Tables.
4. The manuscript should be carefully read for typos (I spotted ypu instead of you for example in section 2.2) and not always past tense in many part was used (i.e. pag 5 is instead of was and pag 8 present instead of presented).
5. Section 2.4 please specify that binreg is a funtion of STATA. Otherwise drop it. Prevalence risk ratio (PRR) instead of PRD.
Change regression model was executed with "regression model was performed".
6. Results: please change older adults with adults aged 45 years and more.
Please change "The prevalence difference, from each predictor reference category, for complex multimorbidity varied from" with "Regarding age, the prevalence difference, from the reference category, 45-59 years, for complex multimorbidity varied from".
Please change "The highest difference of PRD" with "The highest difference of PRD, from the reference category of no formal education".
Please specify what pp stands for and put it inside brackets; otherwise is not clear.
Pag 8 "The mean number of inpatient visits among complex multimorbidity group was 0.20±0.67" please specify that you reported it as (mean±sd)
7. Authors did not answer to one of my comment: why did the authors not test the mean between groups? I doubted the difference may not significant and so this would affect the conclusions.
8. Figure 2 numbers are not correctly visualized. it seems that there is a - instead of a point for decimals. Please edited it.
8. Please rephrase "Another limitation of this study is use of self-rated health as a proxy indicator health related quality of life as SRH can be confounded by sociodemographic and economic status"
Author Response
Reviewer 4
The manuscript has improved, but there are still some important changes that should have been done and the authors did not fully answered my comments.
- The outcome definition reported in the abstract " Complex multimorbidity refers to the co-occurrence of three or more chronic illnesses across >2many body systems" is not in line with the one the authors used later on section 2.3 "Individuals with complex multimorbidity were defined as those who scored at least three out of eleven system-specific chapters". So please clarify it and make some examples of complex comorbid patient.
Author’s response: Thank you so much for this suggestion. We have now understood the reviewer’s point of view w.r.t. the definition of complex multimorbidity and we apologize for not being able to clearly convey the definition.
Please refer to the following published papers:
Vinjerui KH, Bjerkeset O, Bjorngaard JH, Krokstad S, Douglas KA, Sund ER. Socioeconomic inequalities in the prevalence of complex multimorbidity in a Norwegian population: findings from the cross-sectional HUNT study. BMJ open. 2020 Jun 1;10(6):e036851.
Harrison C, Henderson J, Miller G, Britt H. The prevalence of complex multimorbidity in Australia. Australian and New Zealand journal of public health. 2016 Jun;40(3):239-44.
Our definition of non-complex as well as complex multimorbidity is in line with most of the international peer-reviewed published articles. However, in the section 2.3, we have now changed the operational definition so as to make the methods uniform in line with what has been done. Also, the same is reflected in the abstract too.
Operational definition
Complex Multimorbidity: We say at least three out of eleven system specific chapters are aggregated i.e. more than two body system. Each chapter/body system consists of one or more chronic condition (table 1), hence in three chapters (systems) at least three or more chronic conditions have come justifying our definition “complex multimorbidity [8], which is defined as the co-occurrence of three or more chronic conditions affecting more than two body systems in a single individual.”
Non-complex Multimorbidity: Now refer to Figure 1 where out of 30489 respondents having two or more chronic conditions we grouped 10,523 participants as complex based on above criteria. Then, 19,966 who had two or more chronic conditions across less than two body system (as all participants having chronic conditions across 3 or more body system have already been put in complex group leaving only participants with chronic conditions in one or two body system) have been referred as non-complex.
We hope that we are now able to do justice with the reviewer’s expectations. However, if required we are open to further revision.
- The wealth index should be better specified. You should report the level of income (as intervals) that defined each category. Otherwise it is not clear what do you mean with Poorest, Poorer ...
Author’s response: The interval of income is not specified in the dataset, instead LASI has directly provided wealth quintiles variable based on monthly per capita expenditure of the participant. This point has now been mentioned in the manuscript.
- In Table 2 The column Overall (for all the population) should be added otherwise it's not clear and also in text is not clearly reported i.e. "The majority of the participants came from rural areas, had no formal education, and were currently employed" we cannot see it from Table 2 because the two groups presented different results.
Please also change the format of Table 2, to be in line with the other Tables.
Author’s response: We have added an overall column, thus making the reported text in line with the findings. Format of table has been changed as other tables.
- The manuscript should be carefully read for typos (I spotted ypu instead of you for example in section 2.2) and not always past tense in many part was used (i.e. pag 5 is instead of was and pag 8 present instead of presented).
Author’s response: Thank you for pointing this out. We have now thoroughly verified the text and edited all such mistakes.
- Section 2.4 please specify that binreg is a funtion of STATA. Otherwise drop it. Prevalence risk ratio (PRR) instead of PRD.
Change regression model was executed with "regression model was performed".
Author’s response: We have removed binreg; changed PRD to PRR and edited the line "regression model was performed" as suggested by the reviewer.
- Results: please change older adults with adults aged 45 years and more.
Author’s response: Changed as suggested. (This study included 30489 adults aged ≥45 years who had two)
Please change "The prevalence difference, from each predictor reference category, for complex multimorbidity varied from" with "Regarding age, the prevalence difference, from the reference category, 45-59 years, for complex multimorbidity varied from".
Author’s response: Changed as suggested. (Regarding age, the prevalence difference, from the reference category, 45-49 years, for complex multimorbidity varied from [PRD: 14 (11, 17)] percentage)
Please change "The highest difference of PRD" with "The highest difference of PRD, from the reference category of no formal education".
Author’s response: Changed as suggested. (The highest difference of PRD, from the reference category of no formal education was observed)
Please specify what pp stands for and put it inside brackets; otherwise is not clear.
Author’s response: Changed as suggested. (for complex multimorbidity varied from [PRD: 14 (11, 17)] percentage points (pp) for participants)
Pag 8 "The mean number of inpatient visits among complex multimorbidity group was 0.20±0.67" please specify that you reported it as (mean±sd)
Author’s response: Changed as suggested. (The mean number (mean±sd) of inpatient visits among complex multimorbidity group was 0.20±0.67 as compared to 0.13±0.49 among non-complex multimorbidity (supplementary table S3).)
- Authors did not answer to one of my comment: why did the authors not test the mean between groups? I doubted the difference may not significant and so this would affect the conclusions.
Author’s response: The given data on healthcare expenditure is skewed due to which we have used median (IQR) to present the expenses instead of mean. We feel median will be a better measure over mean to present a skewed data. However, as suggested by the reviewer we have also repeated the analysis with mean and applied Mann-Whitney U test to investigate the mean difference. We found OPD costs to be significantly higher among complex multimorbidity group (see table below). However, we still feel median (IQR) to be better measure of presentation of the data. If the reviewer suggests to keep the new below mentioned table with mean, we’ll do so. Additionally, nowhere in the manuscript we have mentioned that the healthcare costs are significantly higher, we just report the numbers to be higher among complex group as suggested by the median data.
Table s#. Healthcare expenditure in US$ on multimorbidity vs. complex multimorbidity.
|
Type of service availed |
Non-complex Multimorbidity Mean (±SD) |
Complex Multimorbidity Mean (±SD) |
p-value (Mann-Whitney U test) |
|
Outpatient visits |
1725.34 (±5738.94) |
2050.98 (±6802.42) |
<0.001 |
|
Inpatient hospitalization |
30463.12 (±66380.88) |
34694.00 (±74921.01) |
0.086 |
*in INR
|
Type of service availed |
Multimorbidity Present Mean (±SD) |
Complex Multimorbidity Present Mean (±SD) |
|
|
Outpatient visits |
24.85 (±82.65) |
29.54 (±97.96) |
<0.001 |
|
Inpatient hospitalization |
438.70 (±955.95) |
499.63 (±1078.93) |
0.086 |
*1 US$=69.44 INR on 31st Dec, 2018
- Figure 2 numbers are not correctly visualized. it seems that there is a - instead of a point for decimals. Please edited it.
Author’s response: Changed as suggested.
- Please rephrase "Another limitation of this study is use of self-rated health as a proxy indicator health related quality of life as SRH can be confounded by sociodemographic and economic status"
Author’s response: Changed as suggested. (We used self-rated health as a proxy indicator of health related quality of life but SRH can be confounded by socio-demographic and economic status of the participants which is another limitation of the study.)
We thank reviewer for their valuable suggestions and time.